# Fabrication and Characterization of a Novel Composite Magnetic Photocatalyst β-Bi_2_O_3_/BiVO_4_/Mn_x_Zn_1−x_Fe_2_O_4_ for Rhodamine B Degradation under Visible Light

**DOI:** 10.3390/nano10040797

**Published:** 2020-04-21

**Authors:** Yong Cheng, Yahan Yang, Zao Jiang, Longjun Xu, Chenglun Liu

**Affiliations:** 1State Key Laboratory of Coal Mine Disaster Dynamics and Control, Chongqing University, Chongqing 400044, China; 20182002019t@cqu.edu.cn (Y.C.); ailungoal@126.com (Y.Y.); jiangzao520@sina.com (Z.J.); xlclj@cqu.edu.cn (C.L.); 2College of Chemistry and Chemical Engineering, Chongqing University, Chongqing 401331, China

**Keywords:** β-Bi_2_O_3_/BiVO_4_/Mn_x_Zn_1−x_Fe_2_O_4_, hydrothermal and calcination method, magnetic photocatalyst, photocatalytic activity, Rhodamine B

## Abstract

β-Bi_2_O_3_/BiVO_4_/Mn_x_Zn_1−x_Fe_2_O_4_ (BV/MZF) composite magnetic photocatalyst was first synthesized using the hydrothermal and calcination method. BV/MZF was a mesoporous material with most probable pore size and specific surface area of 18 nm and 17.84 m^2^/g, respectively. Due to its high saturation magnetization (2.67 emu/g), the BV/MZF composite can be easily separated and recovered from solution under an external magnetic field. The results of photo-decomposition experiments show that the decomposition rate of Rhodamine B (RhB) by BV/MZF can reach 92.6% in 3 h under visible light. After three cycles, BV/MZF can still maintain structural stability and excellent pollutant degradation effect. In addition, analysis of the photocatalytic mechanism of BV/MZF for RhB shows that the p-n heterojunction formed in BV/MZF plays a vital role in its photocatalytic performance. This work has potential application in the future for solving environmental pollution.

## 1. Introduction

Dye wastewater discharge is increasing with the continuous expansion of the textile and dyeing industries, resulting in higher and higher content of organic pollutants in water [1,2]. As organic pollutants are non-biodegradable, highly toxic, and carcinogenic, they have a serious impact on human health and the ecosystem [3,4]. For example, the basic cationic dye, Rhodamine B (Rhodamine B, RhB), has high chroma, high toxicity, teratogenicity, and hard degradation [5,6]. Therefore, the effective treatment of organic dyes in industrial wastewater is becoming an urgent problem. In recent years, photocatalysts were widely used in wastewater treatment due to their green, environmentally friendly, sustainable, and economic advantages [7,8]. Various semiconductor materials were applied to photocatalysts, such as TiO_2_, ZnO, CdS, and BiVO_4_ [9,10,11,12]. Among them, non-toxic BiVO_4_ was widely studied due to its good optical absorption performance, high ionic conductivity, and excellent band gap energy (2.4 eV) [13]. However, BiVO_4_ has a fast electron-hole pairs recombination rate, a narrow visible light response range, and poor photocatalytic activity. Compounding other semiconductor materials is an effective solution for separating photogenerated electron-hole pairs, thereby greatly improving its catalytic activity [14]. In recent research, the photocatalytic activity of composite photocatalysts such as BiFeO_3_/BiVO_4_, M-BiVO_4_/T-BiVO_4_, and TiO_2_/BiVO_4_ was significantly improved [15,16,17]. Nevertheless, the low reuse and high costs caused by the difficult separation restrict the practical application of composite photocatalysts [18].

Conventional photocatalyst supported magnetic substrates are becoming more and more popular. Compared with traditional metallic soft magnetic materials (e.g., Fe_3_O_4_), Mn_x_Zn_1−x_Fe_2_O_4_ (MZF) has the advantages of high permeability, large saturation magnetization, and strong stability [19]. MZF was successfully used in the preparation of magnetic photocatalysts, such as MZF/β-Bi_2_O_3_, MZF/BiOCl, and MZF/α-Bi_2_O_3_ [19,20,21]. These magnetic photocatalysts have high photocatalytic activity and saturation magnetization for easy separation. However, there have been no reports about the synthesis of composite magnetic photocatalyst BV/MZF so far. In addition, it was reported that the preparation process of MZF needs to be calcined at 1200 °C for 3 h, which consumes a large amount of energy and limits the application of MZF in the treatment of environmental pollution. In this research, not only was the preparation of MZF completed by the hydrothermal method at 200 °C for 5 h, but also composite magnetic photocatalyst BV/MZF was compounded efficiently and environmentally by the hydrothermal and calcination method. Moreover, the structure, composition, surface characteristics, magnetic properties, and photocatalytic activity of BV/MZF have been researched. The results demonstrated that it had a fabulous photodegradation effect on RhB and outstanding stability.

## 2. Materials and Methods

Analytical reagents of Bi(NO_3_)_3_·5H_2_O, Na_2_CO_3_, ZnSO_4_·7H_2_O, MnSO_4_·H_2_O, HNO_3_, H_2_SO_4_, HCl, NaHCO_3_, NaOH, NH_4_VO_3_, Fe_2_(SO_4_)_3_, NH_3_·H_2_O, C_28_H_31_ClN_2_O_3_ (RhB), and tartaric acid were utilized as primeval materials. The above were all purchased from Chengdu Kelong Chemical Ltd. (Chengdu, China). All experimental water used was deionized water.

### 2.1. Synthesis of BV/MZF

BiVO_4_, β-Bi_2_O_3_, and MZF, were prepared according to the literatures [22,23,24], respectively. Primarily, 2 mol/L HNO_3_ was used to dissolve 0.01 mol Bi(NO_3_)_3_·5H_2_O by ultrasonic vibration as solution A. Solution B was gained by dissolving 0.02 mol tartaric acid in hot water at 80 °C. Finally, 0.01 mol NH_4_VO_3_ was added to 80 °C hot water and dissolved as solution C. Afterwards, solution B was slowly added dropwise to solution C, and the resulting solution was referred to as solution D. Subsequently, solution A was added dropwise to solution D. After cooling the mixture to room temperature, the pH was adjusted to 7.5 with NH_3_·H_2_O, and BiVO_4_ precursor was formed. Depending on the mass ratio of 10:100 (the theoretical generation value of BiVO_4_), as-prepared MZF was added to the precursor solution, and the reaction was performed in a water-bath at 80 °C for 30 min. Deionized water was used to wash the resulting precipitates five times and dried at 60 °C for 12 h. Lastly, the precipitates were placed in a muffle furnace at 450 °C for 3 h to obtain BiVO_4_/MZF (10 wt%).

The different mass ratios of BV/MZF were obtained according to the following procedures. Firstly, 4 mmol Bi(NO_3_)_3_·5H_2_O was dissolved in 10 mL of 2 mol/L HNO_3_ solution, and the solution was stirred with a mechanical stirrer for 30 min at ambient temperature. Then, the stirred solution was slowly added to 40 mL of 0.6 mol/L Na_2_CO_3_ solution, and the mixture was mechanically stirred at ambient temperature for 2 h, as a precursor solution. Then, the mass ratios of 5, 10, 15, 20, and 25:100 (β-Bi_2_O_3_ generate theoretically), as-synthesized BiVO_4_/MZF, were added to the precursor solution. The resulting mixture was filtered and washed with deionized water. Finally, the filter residues that had been dried at 60 °C for 12 h were roasted in a muffle furnace at 380 °C for 10 min. The synthesis scheme of BV/MZF is displayed in Figure 1.

### 2.2. Characterization

The phase structure of the synthesized samples was first analyzed and proven using an X-ray powder Diffractometer (Shimadzu, XRD-6000, Kyoto, Japan), and their structures were further demonstrated by Fourier transform infrared spectroscopy (FTIR, Perkin-Elmersystem 2000, Thermo Fisher Scientific, Waltham, MA, USA). Field-emission scanning electron microscopy (SEM, Hitachi S-4800, Hitachi, Tokyo, Japan) and a transmission electron microscope (TEM, Tecnai G2F20, FEI, Hillsboro, OR, USA) were used to investigate the morphology of the targeted samples. Energy dispersive X-ray spectrometry (EDX) in TEM mode was employed for elemental analysis. The specific surface area and pore size distribution, magnetic properties, adsorption properties, and surface element states were confirmed by Brunauer−Emmett−Teller (BET, ASAP-2020, Micromeritics, Norcross, GA, USA), vibrating sample magnetometer (VSM 7410, Lake Shore, Carson, CA, USA), ultraviolet-visible diffuse reflectance spectrophotometer (UV-vis, DRS, TU1901, Persee, Beijing, China) and X-ray photoelectron spectrometer (XPS, ESCALAB250Xi, Thermo Fisher Scientific, Waltham, MA, USA), respectively.

### 2.3. Photocatalytic Tests

Under the irradiation of the 300 W xenon lamp (CEL-HXF 300, Beijing CEAULIGHT Co., Ltd., Beijing, China), with a 420 nm cutoff filter was selected as visible light source, the photocatalytic activity of BV/MZF for RhB solutions as simulated dye wastewater was studied. The temperature and pH of the photocatalytic tests were room temperature and natural pH value, respectively. The prepared 100 mg samples were added to 50 mL of the 10 mg/L RhB aqueous solution, and the dark reaction for 30 min reached the adsorption and desorption equilibrium. During the photocatalytic process, the liquid level of RhB and the light source of the xenon lamp were maintained at 20 cm, and the light experiment was performed under magnetic (normal materials) or mechanical (magnetic materials) stirring. After every 30 min, 5 mL the reaction solution was absorbed and centrifuged for 3 min at 4000 rpm. The photodegradation rate of all samples was the average of three experimental results. Eventually, the decomposition efficiency of the target contaminant was calculated by Equation (1):η = (A_0_−A_t_)/A_0_ × 100%,(1)
where η, A_0_, and A_t_ are degradation efficiency, and the equilibrium concentration of the target contaminant before and after visible light irradiation, respectively.

## 3. Results and Discussion

### 3.1. Synthesis Optimization

The effect of BV/MZF with different mass ratios on the treatment of RhB and UV-visible spectrum of optimal samples are shown in Figure 2. As seen from Figure 2a, the photocatalytic efficiency for RhB with BV/MZF (5–25 wt%) composite catalysts are 92.6%, 92.8%, 86.3%, 85.1%, and 56.2%, respectively, after irradiating for 3 h. This result was caused by the lessening of the effective photocatalytic component β-Bi_2_O_3_ in the photocatalyst as the mass of BiVO_4_/Mn_x_Zn_1−x_Fe_2_O_4_ increases. At the same time, the photodegradation rates of BiVO_4_, β-Bi_2_O_3_, and 20 wt% BV/MZF were relatively close, while it improved for 10 wt% BV/MZF samples. In Figure 2b, the absorbance of the maximum peak at 554 nm decreased with time, indicating that RhB continuously degraded, and confirming that the degradation rate of RhB by BV/MZF (10 wt%) was uniform and stable. Ultimately, the BV/MZF (10 wt%) with the best photocatalytic performance was used as the analytical test sample.

### 3.2. Characterization on the Structure and Specific Surface Property

Figure 3 shows the XRD patterns of β-Bi_2_O_3_, BiVO_4_, MZF, and BV/MZF. The characteristic diffraction peak of the tetragonal β-Bi_2_O_3_ is consistent with JCPDS Card No. 27-0050, and the crystal plane indices correspond to (210), (201), (002), (220), (200), (400), (203), (421), (213), (004), and (400). The monoclinic crystal BiVO_4_ samples are in good agreement with JCPDS Card No. 14-0688, and its crystal plane indices are (110), (011), (121), (040), (200), (002), (211), (150), (240), (042), (220), (161), (321), and (123). The MZF crystal plane has characteristic peaks of cubic spinel structure, and the characteristic peaks at 18.2°, 29.9°, 35.2°, 42.6°, 52.9°, 56.3°, and 61.9° correspond to JCPDS Card No. 10-0467. The crystal plane indices are (111), (220), (311), (222), (440), (422), (511), and (440). Notably, BV/MZF samples not only contain characteristic diffraction peaks of (201), (211), (220), and (421) of β-Bi_2_O_3_, but also contain the characteristic peaks of (200), (211), (220), and (161) of BiVO_4_. Furthermore, there is high intensity reflection at 24.5° in BV/MZF, which is caused by the conversion of monoclinic BiVO_4_ to tetragonal BiVO_4_ during the roasting process [25]. However, only the (440) characteristic diffraction peaks of MZF could be found in BV/MZF. This may be owing to the fact that other diffraction peaks in MZF are so weak that they were covered by the diffraction peaks of β-Bi_2_O_3_ and BiVO_4_ [26]. The characteristic peaks of MZF, BiVO_4_, and β-Bi_2_O_3_ were observed in the composite sample, which indicates that BV/MZF was successfully synthesized.

Figure 4 shows the FTIR spectra of MZF, BiVO_4_, β-Bi_2_O_3_, and BV/MZF. The stretching and bending vibration peaks provided by the hydroxyl groups (–OH) from the surface water molecules can be attributed to 3434.0 cm^−1^ and 2359.3 cm^−1^ of MZF, respectively [27]. The bands at 568.0 cm^−1^ and 1399.4 cm^−1^ are the stretching vibration peaks of the zinc–oxygen bond and typical Raman bands of the ferrite bond, respectively [28]. The characteristic absorption peaks at 1387.6 cm^−1^, 851.7 cm^−1^, and 582.8 cm^−1^ are caused by the bending vibration of the bismuth oxygen bond [29,30]. The absorption peak at 468.3 cm^−1^ is closely related to the stretching vibration of the bismuth oxygen bond [24]. The absorption peak at 472.0 cm^−1^ in Figure 4b and at 641.3 cm^−1^ in Figure 4a,b arise from the stretching vibration peak of the bismuth oxygen bond and the stretching vibration peak of the vanadium oxygen bond, respectively. Due to the low MZF content, the characteristic absorption peak of MZF does not appear in BV/MZF (Figure 4a).

The elemental composition and state of the BV/MZF are analyzed by XPS. Figure 5 shows the XPS full spectrum and narrow scan area map of Zn 2p, Mn 2p, Fe 2p, O 1s, Bi 4f, and V 2p. In Figure 5a, the characteristic peaks of Mn, Zn, Fe, Bi, V, O, and C were detected on the surface of BV/MZF, indicating the presence of these elements. This is consistent with the bonds that contained related elements in the FTIR results. The signal of carbon from the apparatus was responsible for the peak observed at C 1s around 284.06 eV. The two major peaks centered at 158.41 eV and 163.71 eV belong to bismuth 4f_5/2_ and bismuth 4f_7/2_, respectively, which means that the bismuth state of BV/MZF is Bi^3+^ cation (Figure 5b). The peaks at 515.86 eV and 523.47 eV appearing in Figure 5c are generally considered to be vanadium 2p_3/2_ and vanadium 2p_1/2_, which means that vanadium is V^5+^ in BV/MZF. In the end, the bond energy at 529.02 eV, 640.20 eV, 1051.30 eV, and 710.77 eV were generated by the signals of O 1s, Mn 2p, Zn 2p, and Fe 2p, respectively (Figure 5d–g). In summary, the production of BV/MZF was confirmed.

Figure 6 reveals the SEM images of β-Bi_2_O_3_, BiVO_4_, MZF, and BV/MZF. From Figure 6a,b, it is obvious that β-Bi_2_O_3_ is composed of flakes, irregular blocks, and particles of different sizes, while BiVO_4_ shows an irregular dumbbell shape [31]. MZF is irregular and aggregated together, as shown in Figure 6c. The massive flaky β-Bi_2_O_3_ can be clearly found through the blue circle in BV/MZF, and the obvious irregular dumbbell-shaped BiVO_4_ can be found through the red circle in Figure 6d. However, the granular MZF could not be observed because of the relatively low content in BV/MZF.

Figure 7 displays the TEM and high-resolution TEM pictures of BV/MZF. Figure 7a shows that many MZF blocks and irregular dumbbell BiVO_4_ are in contact with the flaky β-Bi_2_O_3_. In Figure 7b, the phase with the lattice fringes (0.273 nm and 0.244 nm) could be observed, which corresponded to the (112) and (202) plane of the BiVO_4_ crystal. The lattice fringes of 0.199 nm and 0.317 nm are located on the (321) and (211) crystal planes of β-Bi_2_O_3_, respectively. The lattice space of 0.255 nm agrees well with the (311) plane of MZF. As seen from Figure 7c, the energy-dispersive X-ray spectroscopy (EDX) spectrum result proves the existence of Mn, Zn, O, Bi, and V elements in BV/MZF. It is worth noting that there is a strong Cu signal at 8 KeV, which may be from the TEM grid [32]. Combining the analyses of XRD and XPS, it was confirmed that BV/MZF was successfully prepared.

Figure 8 is the N_2_ adsorption–desorption isotherm of BV/MZF with its corresponding pore size distribution curve (inset). In Figure 8, the N_2_ adsorption–desorption isotherm and hysteresis loop of BV/MZF obtained by the Brunauer isotherm classification method belong to type “IV” and type H3 respectively. Simultaneously, the sample has a mesoporous structure. From the inset in Figure 8, the most likely pore size for BV/MZF is 18 nm. Moreover, the specific surface area of the BV/MZF is 17.84 m^2^/g. Table 1 exhibits the average pore diameter and specific surface area of BiVO_4_, β-Bi_2_O_3_, MZF, and BV/MZF. From Table 1, the pore size (9.49 nm) and specific surface area (17.84 m^2^/g) of BV/MZF are larger than β-Bi_2_O_3_ (6.09 nm and 11.81 m^2^/g) and BiVO_4_ (6.29 nm and 6.00 m^2^/g). The larger surface area of the BV/MZF nanocomposite provides more active sites during the photocatalytic reaction and enhances the photocatalytic activity.

### 3.3. Magnetic Properties

Figure 9 presents the magnetization curves of MZF and BV/MZF nanocomposites tested using VSM technology. The saturation magnetization (Ms) of MZF and BV/MZF are 70.61 emu/g and 2.67 emu/g, respectively. The saturation magnetization of BV/MZF only accounts for 3.78% of MZF, owing to the lower content of magnetic material MZF in BV/MZF. However, the magnetic properties of BV/MZF are sufficient for quickly recovering the photocatalyst from the reaction solution under an external magnetic field. Besides, BV/MZF exhibits the same magnetic properties as MZF, which means that the composite material contains MZF [33]. 

### 3.4. UV-vis DRS

The optical properties of a semiconductor photocatalytic material have an important relationship with its band structure, which is one of the important factors determining its degradation performance. The UV-vis DRS spectra of BV/MZF is depicted in Figure 10. The largest absorption wavelength and forbidden band energy of BV/MZF are about 593 nm and 2.29 eV, respectively. The literature [22] shows that the largest absorption wavelength and forbidden band energy of β-Bi_2_O_3_ was 530 nm and 2.48 eV, respectively. Similarly, according to reference [23], for BiVO_4_ it was 508 nm and 2.51 eV, respectively. It can be seen that the composite material expands the range of absorbed light. Compared with β-Bi_2_O_3_ and BiVO_4_, BV/MZF has a significantly lower forbidden band energy, resulting in the larger the visible light response range of BV/MZF, which is more conducive to photocatalytic reactions.

### 3.5. Stability and Reusability

To further evaluate the stability and reusability of the BV/MZF, three regeneration experiments were performed. At the end of each experiment, the photocatalyst recovered by the external magnet was washed with anhydrous ethanol and deionized water. After drying at 60 °C for 12 h, the next test was performed until the end of the third experiment. The results of three recovery experiments are shown in Figure 11. After photodegradation of RhB for 3 h, the original degradation rate was 92.8%, which only decreased by 7.9% after three cycles of experiments. This outcome shows that the prepared BM/MZF has favorable stability and reusability.

### 3.6. Degradation Mechanism

So as to obtain the possible photocatalytic mechanism, the band position of BV/MZF was first calculated by Equations (2) and (3) [34],
E_CB_ = χ − E^e^ − 0.5 E_g_(2)
E_VB_ = E_CB_ + E_g_(3)
where the implications of χ, E^e^, E_g_, E_VB_, and E_CB_ were the negative values of the ground potential chemical potential, energy of the free electrons of hydrogen (4.5 eV), band gap energy of BV/MZF, potential of VB, and potential of CB. The χ values of BiVO_4_ and β-Bi_2_O_3_ were calculated to be 6.78 eV and 5.90 eV, respectively. The E_VB_ and E_CB_ values of β-Bi_2_O_3_ were calculated to be 2.67 eV and 0.16 eV, respectively. Similarly, the corresponding E_VB_ (3.51 eV) and E_CB_ (1.03 eV) values of BiVO_4_ were obtained. 

The photocatalytic mechanism belonging to BV/MZF is shown in Figure 12. Under visible light, the photon energy absorbed by BV/MZF satisfies the equation: hν ≥ E_g_. Then, the valence band electrons of BV/MZF transit to the conduction band, and eventually produces holes (h^+^) and photo-generated electrons (e^−^). h^+^ and e^−^ moved with the appearance of BV/MZF and reacted accordingly. Dissolved oxygen attached to the outside of BV/MZF traps e^−^ to form superoxide anions (·O_2_^−^), and h^+^ converts the OH^−^ and H_2_O adsorbed on the surface of BV/MZF into hydroxyl radicals (·OH). Organic pollutants decomposed into other products due to the strong oxidative properties of ·O_2_^−^ and ·OH.

The improvement of BV/MZF photocatalytic activity can be due to the following reasons. (1) Composition of p-n heterojunction. The E_CB_ and E_VB_ of n-type BiVO_4_ are more positive and more negative than p-type β-Bi_2_O_3_, respectively [35]. e^−^ of p-type β-Bi_2_O_3_ will migrate to n-type BiVO_4_, and h^+^ of n-type BiVO_4_ will transfer to p-type β-Bi_2_O_3_, thereby constructing a p-n-type heterojunction. Meanwhile, there is an internal electric field inside the BV/MZF system, which will further boost the migration and detachment of photo-generated carriers. (2) As a result of the reduced E_g_ of BV/MZF, the visible light response is enhanced, and the scope of absorbed light is increased. BV/MZF will absorb more incident hν and generate more e^−^ and h^+^, producing more ·O_2_^−^ and ·OH. (3) The stable magnetic field generated by MZF makes e^−^ directional flow, suppresses the recombination of e^−^ and h^+^, extends the e^−^ life, and raises the photocatalytic activity of BV/MZF. (4) Photosensitive degradation of RhB. Due to the low quantum efficiency and low conversion rate of RhB, there could be weak photodegradation during the degradation process [36].

## 4. Conclusions

The composite magnetic photocatalyst BV/MZF was produced by a sample hydrothermal and calcination method and applied for photodegrading RhB. According to the analysis of chemical composition, valence state, and structure, the three-component photocatalyst BV/MZF was successfully compounded. Under visible light irradiation, BV/MZF showed excellent photocatalytic activity. A reasonable photocatalytic mechanism indicates that the p-n type heterojunction formed at the β-Bi_2_O_3_ and BiVO_4_ interface and the stable electric field provided by MZF are helpful for the transfer and separation of photoelectrons. This paper provides a simple and effective method for the synthesis of composite magnetic photocatalysts, and has potential for application in the fields of green chemistry and environmental protection.

## Figures and Tables

**Figure 1 nanomaterials-10-00797-f001:**
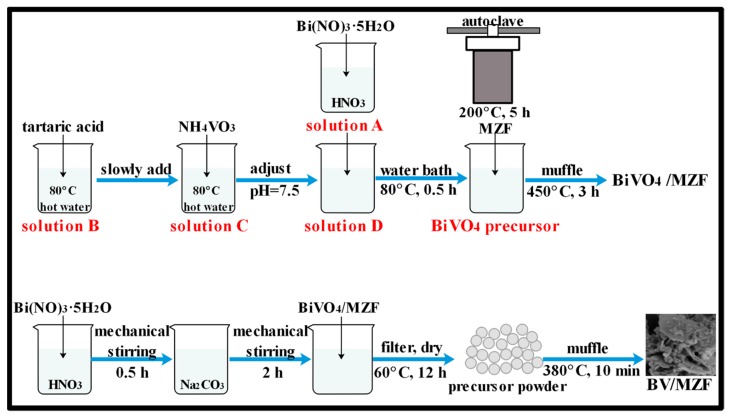
The preparation process of β-Bi_2_O_3_/BiVO_4_/Mn_x_Zn_1−x_Fe_2_O_4_ (BV/MZF) composite.

**Figure 2 nanomaterials-10-00797-f002:**
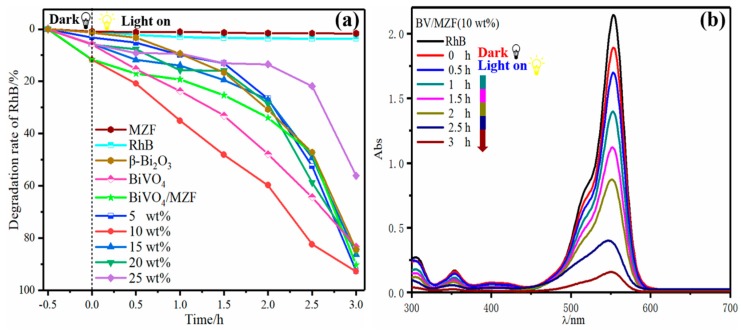
(**a**) The degradation of Rhodamine B (RhB) by BV/MZF (5~25:100). (**b**) Time-dependent UV-vis spectra of RhB solution.

**Figure 3 nanomaterials-10-00797-f003:**
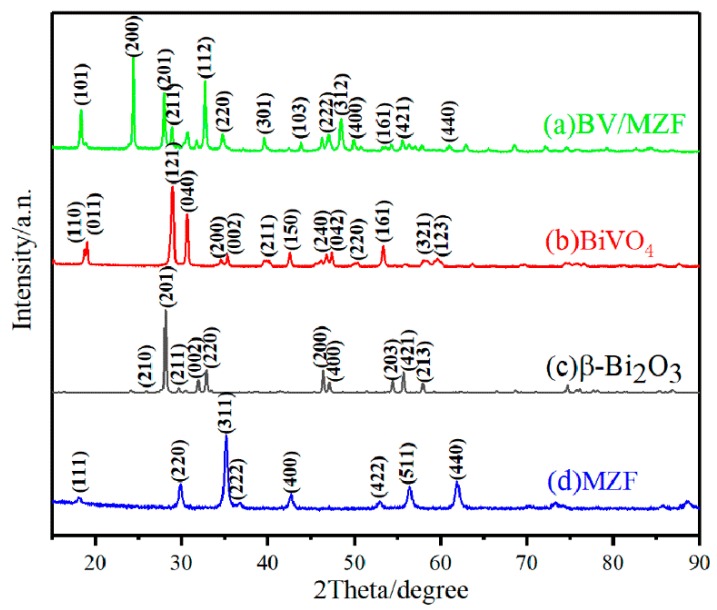
XRD patterns of β-Bi_2_O_3_, BiVO_4_, MZF, and BV/MZF.

**Figure 4 nanomaterials-10-00797-f004:**
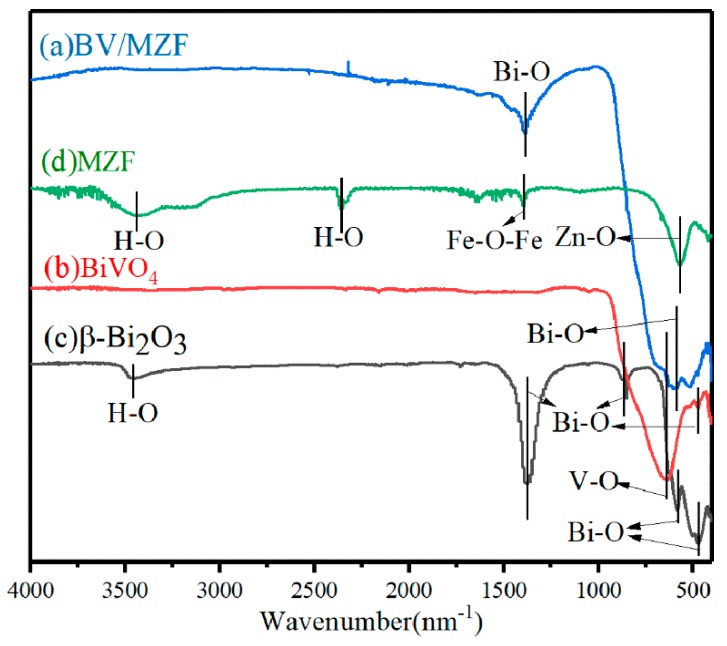
FTIR spectra of β-Bi_2_O_3_, BiVO_4_, MZF, and BV/MZF.

**Figure 5 nanomaterials-10-00797-f005:**
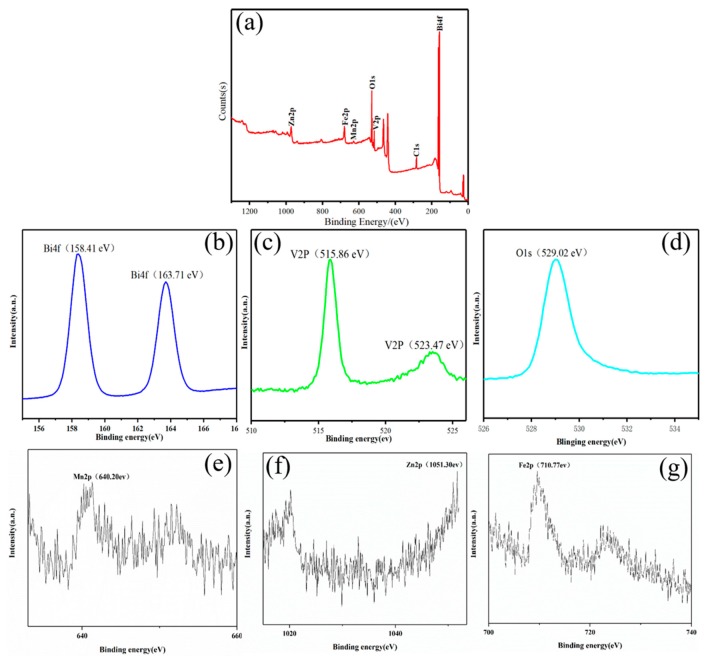
(**a**) BV/MZF complete XPS spectrum. (**b**–**g**) Narrow scan area map of Bi 4 f, V 2p, O 1s, Mn 2p, Zn 2p, and Fe 2p.

**Figure 6 nanomaterials-10-00797-f006:**
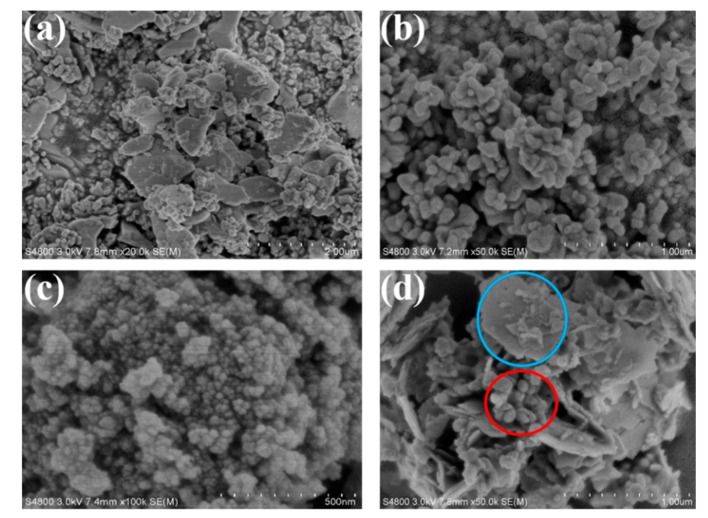
SEM images of (**a**) β-Bi_2_O_3_, (**b**) BiVO_4_, (**c**) MZF, and (**d**) BV/MZF.

**Figure 7 nanomaterials-10-00797-f007:**
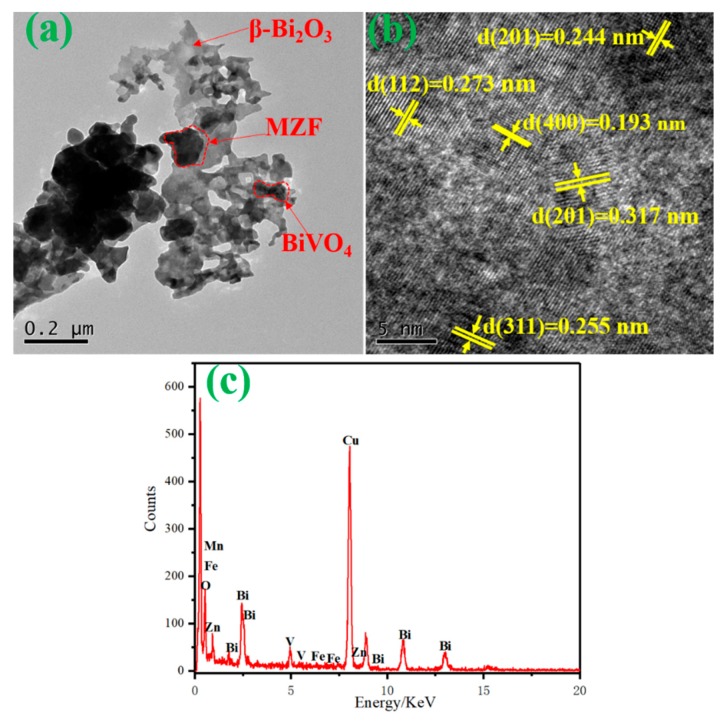
(**a**) TEM diagram of BV/MZF, (**b**) HRTEM diagram of BV/MZF, and (**c**) EDX diagram of BV/MZF.

**Figure 8 nanomaterials-10-00797-f008:**
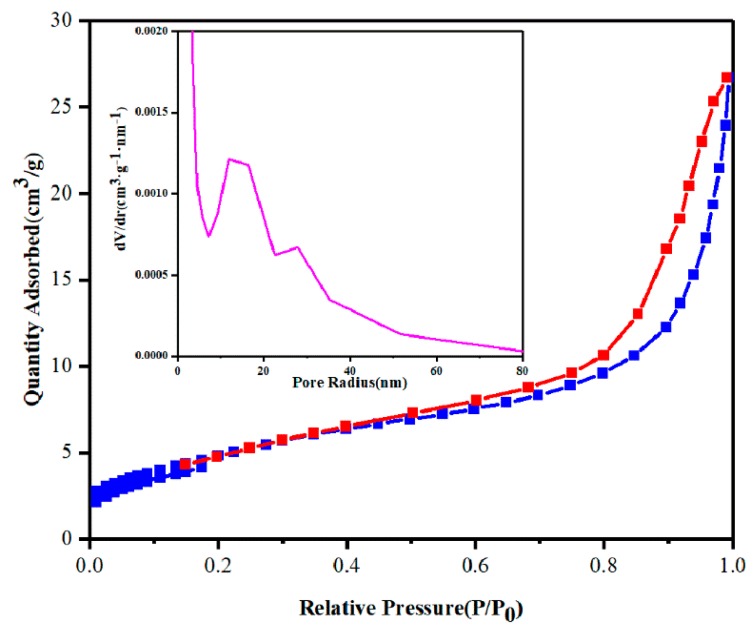
N_2_ adsorption–desorption isotherm of BV/MZF, illustrated with pore size distribution curve.

**Figure 9 nanomaterials-10-00797-f009:**
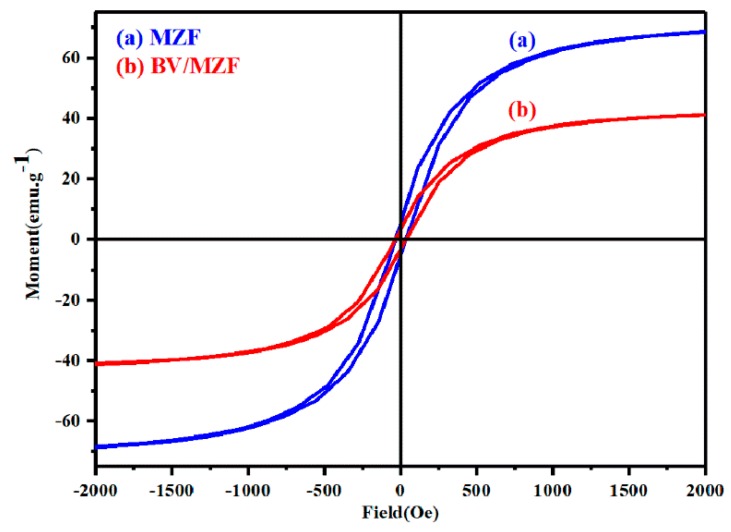
The hysteresis loops of MZF and BV/MZF.

**Figure 10 nanomaterials-10-00797-f010:**
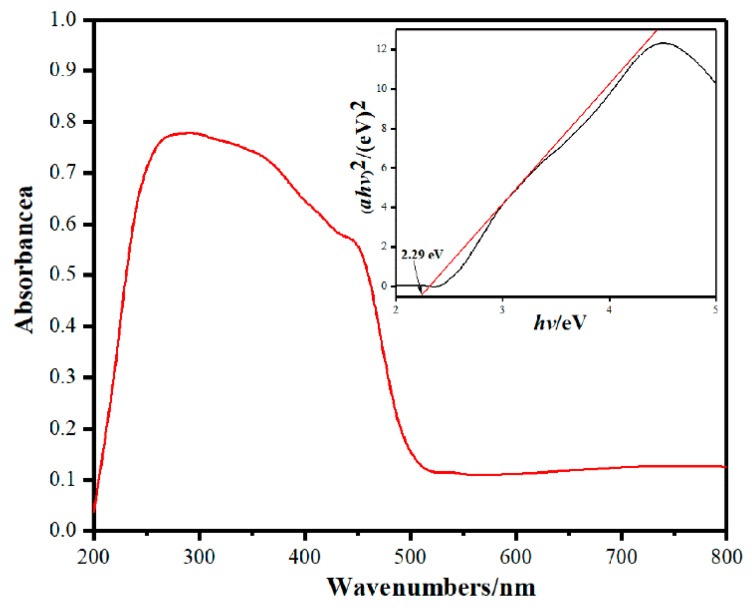
UV-vis DRS spectra and corresponding band gap of BV/MZF.

**Figure 11 nanomaterials-10-00797-f011:**
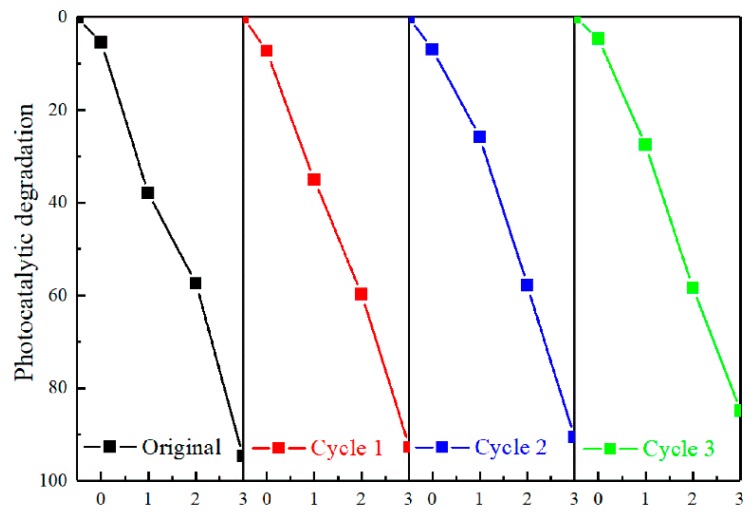
The recovery experiment of RhB degraded by BV/MZF.

**Figure 12 nanomaterials-10-00797-f012:**
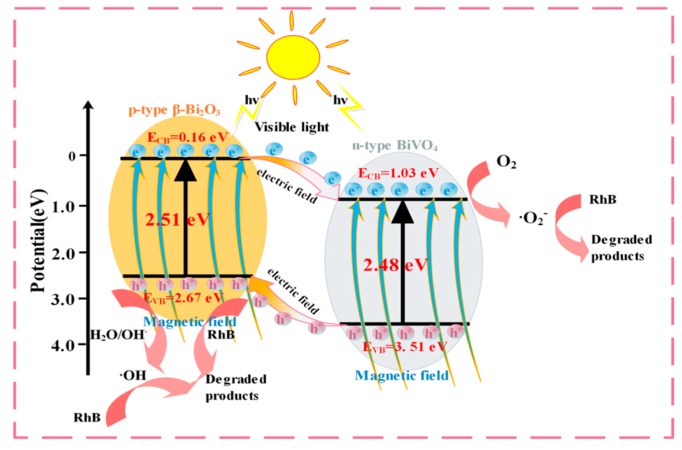
Schematic of the possible reaction mechanism of the photocatalytic procedure.

**Table 1 nanomaterials-10-00797-t001:** BiVO_4_, β-Bi_2_O_3_, MZF, and BV/MZF specific surface area and average aperture.

Samples	BET Surface/m^2^∙g^−1^	Langmuir Surface Area/m^2^∙g^−1^	Pore Size/nm
BiVO_4_	6.00	4.35	6.09
β-Bi_2_O_3_	11.81	11.34	6.29
MZF	53.42	55.60	15.77
BV/MZF	17.84	17.39	9.49

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
