# Peer review of "Fabrication and Characterization of a Novel Composite Magnetic Photocatalyst β-Bi2O3/BiVO4/MnxZn1−xFe2O4 for Rhodamine B Degradation under Visible Light"

_nanomaterials, 2020, doi:10.3390/nano10040797_

Round 1

Reviewer 1 Report

In this manuscript, synthesis, characterization and photocatalytic activity of magnetic composite photocatalyst (β-Bi2O3/BiVO4/MnxZn1-xFe2O4 (BV/MZF) have been studied. The topic of the manuscript is interesting and meets the scope of Nanomaterials. I suggest MAJOR revision. I would like to suggest the authors to take into consideration the following comments and suggestions:

  1. Abstract should be redrafted. Only the most important results of studies should be presented there. For example the sentence - “The as-synthesized samples were characterized by XRD, FTIR, SEM, TEM, XPS, BET, VSM and UV-vis DRS” should be removed.
  2. Authors should try to write the synthesis procedure of the materials in easier way. Now it is difficult to read and understand. Maybe scheme of synthesis can be useful.
  3. Authors used 300 W xenon lamp (CEL-HXF 300, China) as visible light source. What cut-off filter has been applied for obtained vis light? It should be completed.
  4. What about the photoactivity of β-Bi2O3 ? I should be added and compared to other materials.
  5. XRD pattern of BV/MZF should be analyzed once again. Authors stated that BV/MZF is composed of β-Bi2O3 , BiVO4  and MZF. After comparing the XRD patterns of the references samples (β-Bi2O3, BiVO4  and MZF)  with BV/MZF composite, I think that the phase composition of BV/MZF is different. The positions of the reflections assigned to proper phases do not coincide. Moreover, there is high intensity reflection at about 24° (marked as 200). What phase does this reflex come from? This part of manuscript should be corrected.
  6. UV-vis-DRS analysis should be repeated. The quality of spectrum is very low. There is steps in the spectrum. Why?
  7. Does photosensitized decomposition of RhB possible? It should be also included to degradation mechanism.

Author Response

Dear Reviewer,

Thank you for letter and your comments concerning our manuscript entitle “Fabrication and characterization of a novel composite magnetic photocatalyst β-Bi2O3/BiVO4/MnxZn1-xFe2O4 for Rhodamine B degradation under visible light” (ID: nanomterials-760646). These comments are all valuable and very helpful for revising and improving our paper, as well as the important guiding significance to our researches. Based on your comments and requests, we have made extensive modification on the original manuscript which we hope meet with approval. All revisions use the "Track Changes" function in Microsoft Word and are marked in red in the paper, so that changes are easily visible to you. The line numbers and precise changes in the response were obtained under the hide Microsoft Word's revision function. Here, we attached  our response to reviewers' comments in the format of Microsoft Word. Please see the attachment.

Should you have any questions, please contact us without hesitate.

Thank you and best regards.

Sincerely yours,

Tel: +86 13752820583

E-mail: xulj@cqu.edu.cn

Corresponding author: Longjun Xu

Reviewer 2 Report

In this work, the Authors build on their rich set of publications around nanocomposites based around one or more photocatalyists and a magnetic support. The literature has a variety of combinations, including examples by the same group of BVO/MZF and BiO/MZF, although the use of both BVO and BiO with MZF might be novel. Specifically, I think the manuscript would benefit from a clear justification of why this combination is interesting, as the overall performance doesn't look particularly better than reference 19 (sharing some authors), with the exception of the 10 wt% sample. I think the performance of the 10 wt% sample in this case is no fully in line with the other samples, and would need to be repeated / put better into context to validate the main result of this manuscript.

Other, more detailed comments below.

The language should be improved. For example, just from the abstract:
- "hydrothermal-roasting way" is unclear.
"consequences" should be "findings" or similar
"Simultaneously" is not normally used as a conjunction in this way.
"BV/MZF has a saturation magnetization (2.67 emu/g) that can be easily separated and recovered from solution" is not a correct sentence, should be something along the lines of "Due to its high saturation magnetization, the BV/MZF composite can be easily separated and recovered from solution".
"(RhB)s" I don't follow what the "s" here stands for.
I've just pointed out some obvious improvements for the abstract, but the entire manuscript would highly benefit from an extensive language check/rewrite.

- Introduction. In this manuscript and others by the same authors, I struggle to spot the novel aspect. There are so many papers out already with a combination of these nanomaterial, and the only mention to the supposedly novel aspect of this work comes at line 50.

- Section 2.3 - The Xe lamp will have some UV emission. Was any filter used to support the claim of "visible light photocatalysis"?

- Section 3.1 (Photocatalysis results) - what are the confidence intervals on the measurements? How many samples have been tested? It looks like the behaviour is qualitatively very different for the different formulations (with the 10 wt% being a bit of an outlier, since 5 and 15 behave differently). Specifically, since the 10 wt% formulation is the only case in which the performance might be better than BiVO4 or BiVO4/MZF, I think the 10 wt% line should be justified in detail, including the number of runs that have been tested and the spread of the measurements.

- Line 155: The atomic percentages should be reported with a number of digits comparable with the confidence level that is expected from the measurement. Furthermore, the percentages add up to 150% - this should obviously be checked. What percentages are expected from the synthesis of this nominal 10 wt% sample?

- Fig. 5 - SEM: There is no mention of what the stuctures visible as flakes in a), which I would consider to be much clearer evidence of the presence of Bi2O3 in d) than the justification in the text.

- Fig. 6 - EDX data. The peak around 8 keV is probably wrongly attributed to Bi. Bi has L lines at 10.8, 10.7 and M lines at 2.4 (all in keV). If I had to guess, I'd say that the feature at 8 is probably Cu from the TEM grid. Similarly, the Zn peak looks at 8.9 (another Cu peak position) rather than the expected 8.6.
As a side note - the plot for the EDX data looks stretched horizontally. Rescaling the plot, cutting at 20 keV, would make features more easily readable.

- EDX data: How was the spectrum acquired? In bright field TEM mode one expects to see a lot of spurious signal, including Cu (from the grid), Fe (from the lenses) - the Fe signal, already very low, could be coming from the microscope lenses rather than the sample. I'd encourage the authors to use EDX in STEM and show the elemental distribution of the components if the F20 they have access to has STEM-EDX capabilities.

- Fig. 10: The vertical axis should probably be just degradation, not degradation rate.

- Section 3.6: I don't find the explanation for increased efficiency of BO/BVO/MZF over BVO/MZF convincing. If the case that is made here was valid, why do all the compositions, except for the 10 wt%, are worse photocatalysts than plain BVO/MZF? If the explanation was valid, I'd have expected a slight improvement in all cases of BiO addition, with varied results depending on loading.

Author Response

(The authors gave the same response as above.)

Round 2

Reviewer 1 Report

I recommend publishing the revised manuscript in Nanomaterials.

Author Response

Dear Reviewer,

Thanks very much for your kind work and consideration on publication of our paper. On behalf of my co-authors, we would like to express our great appreciation to editor and reviewers.

Thank you and best regards.

Yours sincerely,

Tel: +86 13752820583

E-mail: xulj@cqu.edu.cn

Corresponding author: Longjun Xu

Reviewer 2 Report

I think the current manuscript is greatly improved. Fig 1 is a very useful addition.

line 96: "attached to TEM" is not clear. I'd change it to "EDX in TEM mode".

Fig 7c - it is now clearer and better labelled. I suggest to also label the Cu peak at 8 keV because it's a very strong feature and to a reader the interpretation as spurious signal from the TEM grid might not be obvious.

Author Response

Dear Reviewer,

Thank you for letter and your comments concerning our manuscript entitle “Fabrication and characterization of a novel composite magnetic photocatalyst β-Bi2O3/BiVO4/MnxZn1-xFe2O4 for Rhodamine B degradation under visible light” (ID: nanomterials-760646). These comments are all valuable and very helpful for revising and improving our paper, as well as the important guiding significance to our researches. Based on your comments and requests, we have revised the manuscript again which we hope meet with approval. All revisions use the "Track Changes" function in Microsoft Word and are marked in red in the paper, so that changes are easily visible to you. The line numbers and precise changes in the response were obtained under the hide Microsoft Word's revision function. Here, we attached revised manuscript and our response to reviewers' comments in the format of Microsoft Word. Please see the attachment.

Should you have any questions, please contact us without hesitate.

Thank you and best regards.

Sincerely yours,

Tel: +86 13752820583

E-mail: xulj@cqu.edu.cn

Corresponding author: Longjun Xu
